# Serologic and Genomic Investigation of West Nile Virus in Kosovo

**DOI:** 10.3390/v16010066

**Published:** 2023-12-30

**Authors:** Petra Emmerich, Xhevat Jakupi, Kurtesh Sherifi, Shemsedin Dreshaj, Ariana Kalaveshi, Christoph Hemmer, Donjeta Pllana Hajdari, Ronald von Possel, Dániel Cadar, Alexandru Tomazatos

**Affiliations:** 1Bernhard Nocht Institute for Tropical Medicine, WHO Collaborating Centre for Arbovirus and Hemorrhagic Fever Reference and Research, 20359 Hamburg, Germany; emmerich@bnitm.de (P.E.); ronald_mueller@yahoo.de (R.v.P.); 2Department of Tropical Medicine and Infectious Diseases, Center of Internal Medicine II, University of Rostock, 18057 Rostock, Germany; christoph.hemmer@uni-rostock.de; 3National Institute of Public Health of Kosovo, 10000 Pristina, Kosovo; xhevat.jakupi@uni-pr.edu (X.J.); ariana.kalaveshi@rks-gov.net (A.K.); donjeta.hajdari@rks-gov.net (D.P.H.); 4Faculty of Agriculture and Veterinary, University of Prishtina “Hasan Prishtina”, 10000 Prishtina, Kosovo; kurtesh.sherifi@uni-pr.edu; 5University Clinic of Infectious Diseases, Faculty of Medicine, University of Pristina, 10000 Pristina, Kosovo; shemsedin.dreshaj@uni-pr.edu

**Keywords:** West Nile virus, Kosovo, serology, phylogeography, genomics, epidemiology

## Abstract

The prevalence of West Nile virus (WNV) is increasing across Europe, with cases emerging in previously unaffected countries. Kosovo is situated in a WNV-endemic region where the seroepidemiological data on WNV in humans remains absent. To address this issue, we have conducted a seroepidemiological investigation of 453 randomly selected sera from a hospital in Kosovo, revealing a 1.55% anti-WNV IgG seroprevalence. Comparative and phylogeographic analyses of the WNV genomes obtained by sequencing archived samples from patients with West Nile fever indicate at least two recent and distinct introductions of WNV lineage 2 into Kosovo from neighboring countries. These findings confirm the eco-epidemiological status of WNV in southeast Europe, where long- and short-range dispersion of lineage 2 strains contributes to a wider circulation via central Europe. Our results suggest an increasing risk for WNV spreading in Kosovo, underscoring the need for an integrated national surveillance program targeting vectors and avian populations for early epidemic detection, as well as the screening of blood donors to gauge the impact of virus circulation on the human population.

## 1. Introduction

West Nile virus is an arthropod-borne flavivirus with an enzootic cycle maintained primarily by *Culex* mosquitoes and susceptible bird hosts. Changes in host ecology and vector feeding behavior often result in spillover to humans or equids, which, despite being epidemiological “dead-ends”, can develop severe illness. Since its discovery in 1937 in Uganda [1], and the first serological confirmation of European cases in 1958 in Albania [2], WNV has been established as an endemo-epidemic virus in southern, central, and western Europe [3]. Unusually high temperatures in recent years have contributed to the northward geographical expansion of WNV in Europe [4,5,6]. In parallel, a marked increase in the numbers of human and equid infections was reported, particularly in central and southeast Europe [3]. In Albania, 22% of tested horses were WNV-seropositive [7], while in Kosovo, a smaller proportion of birds (2%) and horses (10.37%) presented WNV-specific IgG [8]. Although most human infections are asymptomatic, around 20% result in mild flu-like symptoms (e.g., mild fever, headache) and remain largely undiagnosed. Approximately 1% of WNV infections can lead to West Nile neurodegenerative disease (WNND) with encephalitis, meningitis or polio-like paralysis, where increased mortality rates are observed in high-risk groups [1,9]. The expansion of its geographical range in Europe also increases public health risks associated with organ donation and transfusion [10]. Since its initial detection in Hungary in 2004 [11], lineage 2 strains have been the predominant cause of WNV infection in Europe, replacing WNV lineage 1 in endemic areas previously affected by outbreaks (e.g., Romania [12,13]). However, co-circulation still occurs in Italy [14,15] and France [16]. Lineage 2 has shown spatially structured phylogenetic clades in south, southeast, and central Europe, with specific mutations detected in endemic areas hosting major bird migration hubs in Italy, Romania, and Greece [17,18,19]. This suggests a dynamic circulation sustained by the local evolution and transit of migratory bird hosts between Africa and Eurasia. Over the past few years, there has been an increasing number of serological and genomic studies on WNV epidemiology in southeast Europe [18,20,21,22,23,24]. However, there is no information about the impact of WNV on human health in Kosovo, while only scarce data are available from surveillance of animal hosts [7,8]. The present study aims to bridge this gap by conducting a serological and genomic investigation of WNV in humans from Kosovo. First, we retrospectively screened sera for the presence of WNV-specific antibodies in former patients with no signs of West Nile disease. In a second analysis, we explored the origin of WNV introduction in Kosovo and its broader patterns of dispersion in the region. Overall, our results provide useful insights into the WNV dynamics in the region and associated risk for the human population. 

## 2. Materials and Methods

### 2.1. WNV Serology

A retrospective serological survey was conducted on 453 randomly selected patient sera collected between 2012 and 2018 for reasons unrelated to WNV infection or central nervous system disease. These sera, together with cerebrospinal fluid (CSF) samples of four WNND cases (2018), were screened via ELISA for presence of WNV IgG and IgM antibodies (Anti-WNV ELISA IgG and Anti-WNV ELISA IgM, (Euroimmun, Lübeck, Germany), and WNV IgM Capture DxSelect/WNV IgG DxSelect (Focus Diagnostics, CA, USA) according to the manufacturer’s instructions. 

Cross-reaction between WNV and other pathogenic flaviviruses was evaluated by subsequent testing with an in-house immunofluorescence assay (IIFT) for the detection of WNV, Usutu virus (USUV), tick-borne encephalitis virus (TBEV), Japanese encephalitis virus (JEV), and dengue virus (DENV). Reactions were considered specific when the titer of a certain virus was at least 4 times higher than for any other virus. Briefly, infected Vero E6 cells (ATCC CRL-1008) were spotted on glass slides, air dried, and fixed in acetone. Serial dilutions of patient sera were incubated on slides for 1 h at 37 °C. The slides were washed twice with PBS. IgM and IgG antibodies were detected via indirect immunofluorescence using anti-human IgG or anti-human IgM (Medac, Hamburg, Germany) FITC-labeled secondary antibodies after incubation at 37 °C for 25 min. Titers of 1:20 or higher were considered positive. 

Confirmation was conducted through virus neutralization tests (VNT). These are considered the gold-standard method for accurately determining the presence of WNV or any potential cross-reactivity in serological diagnostics. The presence of specific neutralizing antibodies against WNV was conducted on Vero E6 cells using WNV strain NY99 and USUV strain 1477 [8]. Titers of 1:20 or higher were considered positive. Data were analyzed with SPSS v.23, and statistical significance was considered when *p* < 0.05. 

### 2.2. Whole Genome Sequencing

Samples of serum and urine collected from four WNND patients displaying clinical signs and treated in 2018 were confirmed via WNV qRT-PCR then sequenced using a metagenomic next-generation sequencing (NGS) approach. These patients had no history of travel before the onset of illness and were from three distinct regions in Kosovo (Gjilan, Drenas, Pristina). The samples were subjected to a filtration process using a 0.45-μm filter (Millipore, Darmstadt, Germany) for the removal of cellular and bacterial debris. After filtration, 250 µL of the remaining sample was treated with a cocktail of nucleases including Turbo DNase (Ambion, Carlsbad, CA, USA), Baseline-ZERO (Epicenter, Madison, WI, USA), and Benzonase (Novagen, San Diego, CA, USA) for viral particle enrichment. Extracted viral RNA was processed using the QIAmp Viral RNA Mini kit (Qiagen, Hilden, Germany) and sequenced following a previously described protocol [25]. Briefly, for reverse transcription and cDNA synthesis, 10 μL of the extracted RNA was mixed with 100 pmol of a random primer and incubated at 72 °C for 2 minutes. First-strand synthesis was then carried out in a reaction mix containing SuperScript™ III reverse transcriptase, 10 mM dNTPs (Qiagen, Hilden, Germany), 5× first-strand extension buffer, and 10 mM dithiothreitol. This mixture was incubated at 25 °C for 10 minutes, followed by incubation at 50 °C for 1 hour and 70 °C for 15 minutes. The second-strand reaction was conducted using 20 pmol of a random primer, 2 µL of 10× Klenow buffer, and 5U of Klenow Fragment (New England Biolabs, USA) at 37 °C for 1 hour. The obtained dsDNA was then used for library preparation using the QIAseq FX DNA Library Kit (Qiagen, Hilden, Germany). The normalized libraries were sequenced using 150-cycle (2 × 75 bp paired-end) reagent kit (Illumina, San Diego, CA, USA) on a NextSeq550 platform (Illumina, San Diego, CA, USA). The generated WNV genomes were deposited in GenBank under the accession numbers MZ190464-MZ190467. 

### 2.3. Genome Characterization and Phylogeography of WNV

Complete nucleotide sequences of WNV (European lineage 2 strains) with known sampling time (year) and geographical origin (country) were retrieved from Genbank. The dataset was curated in Geneious Prime v2021.1 (Dotmatics, Hertfordshire, UK) and aligned with the MAFFT algorithm. Using a maximum likelihood phylogeny constructed with IQ-TREE 1.6.12 [26], we assessed the temporal structure (i.e., clock-like evolution) in TempEst [27] by conducting a regression analysis of the root-to-tip branch distances as a function of sampling year. To evaluate the spatiotemporal dynamics of WNV and the time to the most recent common ancestor (tMRCA) of WNV in Kosovo, we performed a Bayesian phylogenetic analysis with BEAST v1.10.4 [16], employing a flexible demographic model (Gaussian Random Field) with a HKY+Γ4 nucleotide substitution model and a relaxed uncorrelated lognormal molecular clock following model selection via path sampling/stepping stone [28]. The Markov Chain Monte Carlo (MCMC) chain lengths were set to run for 10^8^ generations, discarding the initial 10% (burn-in) with subsampling every 10^4^ iterations to achieve convergence. Tracer 1.7.1 [29] was used to assess the convergence of the analysis, and the maximum clade credibility (MCC) trees were visualized with FigTree 1.4.3. To investigate the hypothesis of periodic introductions of WNV into Kosovo, we performed a discrete phylogeographic analysis. The Bayesian stochastic search variable (BSSV) algorithm was employed for estimation of non-zero rates of virus migration between locations [30]. SpreadD3 v. 0.9.7.1 [31] was employed to perform the BSSV analysis, generating both the Bayes factor (BF >3) and posterior probability (PP). These were then used to examine and assess the statistically significant epidemiological links between discrete sampling locations. 

## 3. Results

### 3.1. Sero-Epidemiological Survey 

Flavivirus cross-reaction was observed for CSF samples from WNND patients, with the highest titers of anti-WNV antibodies detected in each of them. Additionally, these samples were also confirmed as WNV-positive by VNT. Among the 453 serum samples, seven of them showed presences of anti-WNV IgG (1.55%), two of which also contained anti-WNV IgM antibodies. Of these seropositive samples, six were confirmed by VNT (1.32%, CI 0.75–3.15) (Table 1).

### 3.2. Genomic and Phylogeographic Analysis

The sequenced WNV genomes from Kosovo showed a low degree of genetic variation (0.1–0.8% for nucleotide sequence and 0–0.6% for the amino acid, respectively). Although we generated a small number of genomes, the same pattern of variation was observed for the southeast European clade (SEEC) dataset (n = 47, 0.1–1% for nucleotides and 0.1−0.6% for amino acid sequences). The comparative analysis of SEEC polyproteins revealed a mutational profile that largely reflects the MCC tree topology. Most non-synonymous substitutions found in Kosovar WNV strains were unique (17 of 22 sites), whilst amino acid mutations shared with the rest of SEEC were present on both structural and non-structural proteins (Figure 1). Moreover, multiple non-synonymous mutations, primarily found within the nonstructural genes, showcase distinct geographical patterns specific to the SEEC genomes (Figure 1).

To explore the evolutionary relationship between the strains circulating regionally and the potential source of WNV introduction in Kosovo, a Bayesian MCMC sampling method was employed. The MCC phylogeny of the European WNV lineage 2 dataset indicated that the WNV strains from Kosovo were grouped into two distinct and well-supported subclades within the SEEC (Figure 2). 

The results suggest that the circulation of WNV in Kosovo was triggered by at least two independent introductions from neighboring countries, one of which (Serbia) is a main hub of WNV dissemination in southeast and central Europe (Figure 2 and Figure 3). The estimation of the tMRCA concerning the Kosovar WNV strains suggests a recent emergence. It is probable that these strains were introduced into Kosovo around 2010–2012. Among the BF estimates, the most robust epidemiological connections were identified between Kosovo and Bulgaria (Bayes factor 4.3), Bulgaria and Kosovo (Bayes factor 4.9), and Serbia and Kosovo (Bayes factor 4.4) (Figure 3). The initial introduction and migration event of WNV lineage 2 into Kosovo was traced back to an ancestor that existed in Serbia around 2012 (95% HPD 2011–2014; posterior probability = 0.98), followed by a second introduction, likely a descendant of an ancestor that probably circulated in Bulgaria, estimated to have emerged around 2010 (95% HPD 2009–2013; posterior probability = 1).

## 4. Discussion

After the 1996 Romanian West Nile fever epidemic caused by WNV lineage 1 (over 800 hospital admissions with suspected nervous system infections, of which 393 cases with confirmed neurologic diagnosis) [12], the introduction of lineage 2 via Hungary initially resulted in sporadic cases of neurological disease in birds of prey and mammals. A swift development of disease foci followed in 2008, afflicting birds of prey, horses, and humans across Hungary and eastern Austria [32]. In this period, virus endemisation caused sporadic cases or smaller outbreaks in humans and horses from Bulgaria, Greece, Romania, and Serbia. The region has been experiencing annual increases of West Nile fever cases since 2010, a period in which countries such as North Macedonia, Bosnia and Herzegovina, Montenegro, and Bulgaria have reported the first cases of WNV infection [9]. Notable WNV occurrence has been observed across central and western Europe since 2018, when spectacular increases in infection occurred in birds, horses, and humans [33,34], with a concomitant northwestward expansion into the Netherlands and Germany [4,6]. Following the 2010 upsurge, the number of cases reported in 2018 indicated the largest outbreak to date, with more cases in that season than cumulatively reported between 2010 and 2017 [33]. In southeast Europe, this epidemiological trend was particularly visible in Greece, where, in 2018, the outbreaks started earlier than usual (week 22), with 317 autochthonous cases reported, of which 243 progressed to WNND (23% increase versus 2010), and there were 51 fatalities [35]. We found WNV to have a wider circulation in Kosovo, a fact supported by recent reports of seropositive horses and poultry from multiple areas of the country [8]. A study conducted in Bulgaria on a cohort of 1451 residents (28 districts) reported a seroprevalence of 1.5% for WNV-specific IgG antibodies [36]. These findings are similar to the seroprevalence herein reported (1.55%), but markedly lower than the levels found in Serbia (4%) [20] or Greece (17.3%) [37]. Furthermore, endemic circulation of WNV in Bulgaria was evidenced in 2018 by the detection of viral RNA in Culex mosquitoes and humans [23].

An important aspect of arbovirus epidemiology is silent enzootic circulation when the virus circulates between vectors and animal reservoir hosts without causing overt disease. However, virus transmission in this context has a high potential for rapid amplification to epidemic levels in favorable ecoclimatic conditions [38,39]. The high incidence of WNV infection in south and southeast Europe is known to result from a combination of these factors [40]. Due to infrequent symptomatic manifestations in humans and the possibility of self-limited meningitis in young and immunocompetent individuals [41], the real number of infections is much higher than the one reported [42]. Although laboratory testing is aimed specifically at WNV when infection is suspected, the occurrence of WNND is still considered rare, especially when considering the initial signs of infection. Thus, diagnosis may encounter difficulties, requiring a combination of molecular and serological assays for case confirmation. A particular obstacle in diagnosis of WNV infection is the antigenic cross-reactivity of flaviviruses from the Japanese encephalitis serogroup [43]. Due to high similarity with respect to eco-epidemiology, antigenicity, and the clinical spectrum, USUV infections may significantly hamper diagnosis in WNV-endemic areas. Since 2009, human cases of severe encephalitis due to USUV infection have been recorded in south and central Europe, including in Hungary [44]. Although our seroepidemiological investigation did not detect USUV in Kosovo, the virus was recently identified in mosquitoes from Romania [45], indicating that it is circulating in the region. Thus, in addition to consistent surveillance efforts, the awareness of physicians is also a critical aspect, as highlighted by the 2018 WNV outbreak from Greece [35]. Moreover, seroprevalence investigations conducted in Italy, where both WNV and USUV are circulating, revealed a higher prevalence of USUV-specific antibodies compared to antibodies against WNV. This suggests that USUV infections, for the most part, are asymptomatic [46,47,48]. 

As with most studies attempting to reveal patterns of WNV dispersion in this region, our analysis is limited by the dearth of complete or near-complete genome sequences available from southeast Europe. Thus, some epidemiological links may be obscured by the absence of data, especially for the years and locations of the initial lineage 2 spread (e.g., from Hungary to its neighbors [32]). Despite the limited number of WNV genomes from Kosovo, the low genetic diversity observed among them closely resembles the overall level detected within the SEEC. However, purifying selection acting during transmission balances the high mutation rates specific to RNA viruses (particularly due to the modes of transmission and infection of arboviruses). This observation aligns with the geographic and temporal structure observed within the main European WNV lineage 2 clades, reflecting the evolutionary pattern seen in USUV across Europe [49]. Mutations at positions T726A (E), K883N (NS1), and V2386M (NS4b) have been associated with the emergence of southeast European subclade 2, while L114M (pr) and K1720R (NS3) are exclusive to the southeast European subclade 1. Comparable trends of convergent evolution have been noted in the case of WNV. The virus’ adaptation within vector and vertebrate hosts due to local overwintering or reintroduction may significantly influence the spatial spread and establishment of WNV. It is worth noting the distinctive mutations identified in the WNV strains from Kosovo. However, the precise impact of these mutations on the virus’s fitness remains uncertain. Future insights into fitness and pathogenicity may emerge with additional genomic data, as well as in vitro and in vivo experiments using Kosovo-derived strains. 

Bird migration along the Afro-Eurasian flyway network was posited as a mechanism contributing to WNV introduction in Europe and the Middle East [50]. Long-range migration along this flyway network may have a role in WNV introduction in Europe during host stop-overs at important migration hubs along major flight corridors (e.g., Italy, Greece, Eastern Mediterranean). Nevertheless, considering the focal and endemic character of WNV transmission in Europe, it is likely that the virus relies mainly on local movement of long-range migratory and/or resident birds (especially in dense, urban settings) [33]. Recent examples suggesting such a dispersion pattern have been provided by its recent emergence in Germany [6]. The overwintering of competent vectors is probably another important factor which requires elucidation in endemic regions of Europe. Considering the intensity of WNV circulation in southeast Europe, it is unsurprising that the virus has been circulating in Kosovo for at least a decade. Nonetheless, the results of our phylogeographic analysis require caution in their interpretation due to the paucity of WNV genomes in southeast Europe and the ensuing sampling bias. We may, however, speculate that WNV circulation in Kosovo is comparable in its intensity to that of neighboring countries (e.g., Serbia, Hungary, Greece), where higher prevalence is detected by integrated surveillance [51,52,53]. 

This study offers initial insights into serological and phylogeographic patterns of WNV circulation in Kosovo. It underscores the significant risk posed by human WNV infections. As a response, the surveillance of mosquitoes, birds, and incidental hosts (humans, equids, and dogs) should be implemented as a public health priority. Moreover, our study emphasizes the necessity of generating a larger dataset of complete genomes which will enable a more detailed understanding of WNV dispersion and evolution.

## Figures and Tables

**Figure 1 viruses-16-00066-f001:**
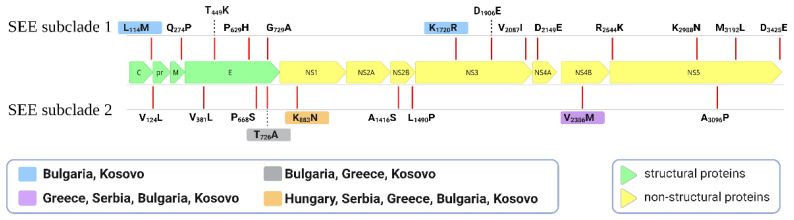
Schematic illustration depicting the WNV genome along with the positions of amino acid mutations in strains from Kosovo and in the context of the lineage 2 southeast European clade (SEEC). Colors indicate identical amino acids shared at a given genome position by Kosovar WNV genomes and sequences from other countries. Mutations without a color box are unique for Kosovar strains relative to the rest of the genomes in this clade.

**Figure 2 viruses-16-00066-f002:**
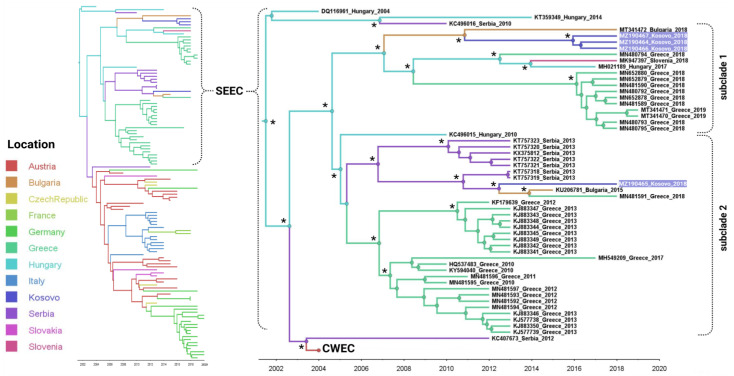
The Bayesian maximum clade credibility (MCC) tree has been constructed to depict the time-scaled phylogeny of West Nile virus (WNV) lineage 2, based on complete genome sequences, which includes the southeast European subclade (SEEC). The colored tree branches indicate the most likely geographic location of their descendant nodes. The time axis below the tree represents years before the last sampling time. Bayesian posterior probabilities (≥90%) and maximum likelihood bootstrap support (≥80%) are denoted by asterisks. CWEC: central west European clade.

**Figure 3 viruses-16-00066-f003:**
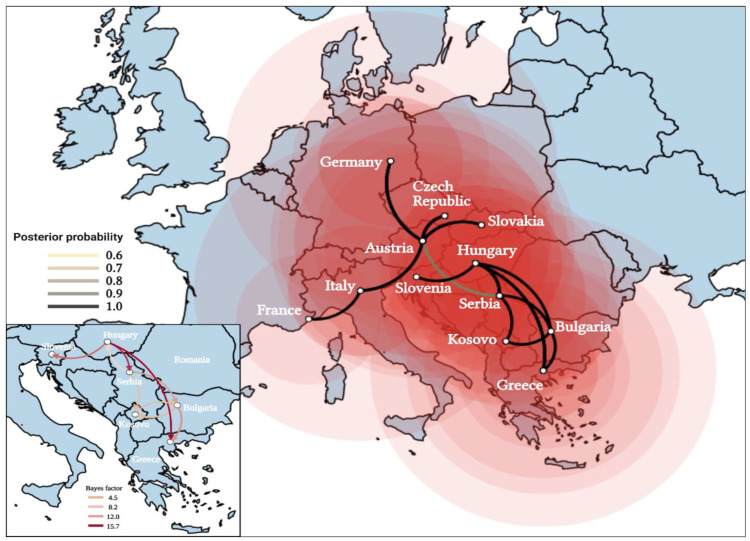
The spread pattern of the European clade of WNV lineage 2, including the origin of the Kosovar WNV. Directed lines between locations represent connections between source and target countries. Circles on the map denote discrete geographical locations of viral strains, aligning with branches in the MCC tree, indicating the relevant location transitions. Circle diameters for each location are proportional to the square root of the number of MCC branches maintaining a specific location state at each timepoint. The inset map displays the migration patterns of WNV between Kosovo and its neighbors. Arrows represent viral migration between the countries, with the color of the arrows being proportional to the strength of the transmission rate, as indicated by the Bayes factor (BF).

**Table 1 viruses-16-00066-t001:** Serological results obtained by IIFT and VNT on the WNV ELISA-reactive archived serum samples.

Sample	IIFT WNV		IIFT USUV		IIFT DENV		IIFT TBEV		VNT	
	IgG	IgM	IgG	IgM	IgG	IgM	IgG	IgM	WNV	USUV
G15-A	1:40	neg	neg	neg	neg	neg	neg	neg	1:80	neg
G15-B	1:640	1:40	neg	neg	1:20	neg	1:20	neg	n.t.	n.t.
G15-C	1:80	neg	neg	neg	neg	neg	neg	neg	1:320	neg
G15-D	1:640	neg	neg	neg	neg	neg	neg	neg	1:640	neg
G15-E	1:2560	1:640	1:20	neg	1:40	neg	1:10	neg	1:2560	1:20
G15-F	1:640	neg	1:20	neg	1:40	neg	neg	neg	1:640	1:40
G15-G	1:320	neg	neg	neg	1:40	neg	neg	neg	1:320	neg

WNV, West Nile virus; USUV, Usutu virus; DENV, dengue virus; TBEV, tick-borne encephalitis virus; IIFT, indirect immunofluorescence testing; VNT, virus neutralization test; IgG, immunoglobulin G; IgM, immunoglobulin M; n.t., not tested; neg, negative (titers < 1:20).

## Data Availability

The generated WNV genomes were deposited in GenBank under the accession numbers MZ190464-MZ190467.

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
