# Peer review of "Serologic and Genomic Investigation of West Nile Virus in Kosovo"

_viruses, 2023, doi:10.3390/v16010066_

Round 1

Reviewer 1 Report

Comments and Suggestions for Authors

In this manuscript the authors present the results of a serological screening of patients in Kosovo and analyze the genomes of four West Nile virus infected cases. The English needs to be revised to make the manuscript more easy to read.

Specific comments:

Please revise lines 20-25 to make the message more clear.

Line 38. Delete The and change to West Nile virus is a mosquito...

Line 44. Delete itself.

Line 50. 20% of the cases include mild symptoms that may include fever and headache but the term West Nile fever is used usually to patients with more severe symptoms...note that most of these 20% cases remains undiagnosed.

Line 55. Please indicate that lineage 2 predominates in recent years because lineage 1 circulates in Western Europe and was the predominant strain for many years in Italy or for example caused the 1996 outbreak in Romania that you mention.

Line 64. Please rewrite. Indicate more clearly that you did a screening for WNV antibody detection using randomly choosen sera from patients going to the hospital with not West Nile Disease simptoms. I assume that these sera are not from blood donors...please clarify.

Line 67-68. It is not clear to me why getting genomic data is important for implementation of an integrated surveillance system. Please explain or delete. Genomes are important for understanding the population dynamics and dispersal of the virus but for surveillance a simple real time pcr is enough, what is important is rapid analyses to detect the virus in the mosquitoes before the first human cases and before the risk of transmission through blood donation gets important.

Please explain the abbreviations at first use. What is CSF (line 75). Note for example that in Figure 2 you explain the meaning of SEEC, but you are already using the term in Figure 1 without explaining what it is, or you use the term in line 170 without explaining before what it means. Please check if you can delete some abbreviations because the manuscript is full of abbreviations that often are only used one or two times.

Line 89. Explain how you dealt with cross-reactions. The usual approach is to consider only as specific those reactions with titters at least 4 times stronger than for any other virus tested.

Line 132-135. Please check these numbers. It seems estrange that you us 108 generations, discard 10% and subsample every 104 iterations...this gives a single value, or maybe I am missing something?

Line 173. Distinct from what? please explain.

Line 181. Please consider changing relatives by sequences.

Line 214. Add - between Bulgaria and Kosovo.

Line 269-277. Some parts seem more adequate for the introduction.

No discussion is done of the serological results, you comment that Usutu is very prevalent in Italy, but I think that you must say specifically that your analyses have only found indication of WNV exposure in humans and despite Usutu is increasingly being reported in Europe you have not found serological evidence of its circulation in Kosovo.

Line 299. Why you are comparing your results for Usutu? please explain the reasons.

Comments on the Quality of English Language

I have added comments on the English in the previous section. Please, specifically check the abstract and the introduction.

Reviewer 2 Report

Comments and Suggestions for Authors

I am concerned by the fact that the authors did not provide any information about the ethical approval. I highly recommend that you seek help with English editing.

Comments on the Quality of English Language

I have an impression that English is not the authors' first language. While this should not disadvantage them, I highly recommend that the authors seek help with English editing. The manuscript should not be be published in its current status.
